# Pharyngeal Communities and Antimicrobial Resistance in Pangolins in Gabon

Johanna P. Wiethoff,[a,b,c] Sarah Sandmann,[d] Tom Theiler,[e] Chimene Nze Nkogue,[f] Etienne-François Akomo-Okoue,[f] Julian Varghese,[d] Andrea Kreidenweiss,[b,c] Alexander Mellmann,[g] Bertrand Lell,[a,b,c] Ayôla A. Adegnika,[a,b,c] Jana Held,[b,c] Frieder Schaumburg[a,e]

[a]Centre de Recherches Médicales de Lambaréné, Lambaréné, Gabon
[b]Institute of Tropical Medicine, University of Tübingen, Tübingen, Germany
[c]German Center for Infection Research (DZIF), Tübingen, Germany
[d]Institute of Medical Informatics, University of Münster, Münster, Germany
[e]Institute of Medical Microbiology, University of Münster, Münster, Germany
[f]Institut de Recherches sur l'Ecologie Tropicale, Libreville, Gabon
[g]Institute for Hygiene, University of Münster, Münster, Germany

**ABSTRACT** Wildlife can be a reservoir and source of zoonotic pathogens for humans. For instance, pangolins were considered one of the potential animal reservoirs of SARS-CoV-2. The aim of this study was to assess the prevalence of antimicrobial-resistant species (e.g., extended-spectrum $\beta$-lactamase [ESBL]-producing *Enterobacterales*) and *Staphylococcus aureus*-related complex and to describe the bacterial community in wild Gabonese pangolins. The pharyngeal colonization of pangolins sold in Gabon ($n = 89$, 2021 to 2022) was analyzed using culture media selective for ESBL-producing *Enterobacterales*, *S. aureus*-related complex, Gram-positive bacteria and nonfermenters. Phylogenetic analyses of ESBL-producing *Enterobacterales* was done using core-genome multilocus sequence typing (cgMLST) and compared with publicly available genomes. Patterns of cooccurring species were detected by network analysis. Of the 439 bacterial isolates, the majority of species belonged to the genus *Pseudomonas* ($n = 170$), followed by *Stenotrophomonas* ($n = 113$) and *Achromobacter* ($n = 37$). Three *Klebsiella pneumoniae* isolates and one *Escherichia coli* isolate were ESBL-producers, which clustered with human isolates from Nigeria (MLST sequence type 1788 [ST1788]) and Gabon (ST38), respectively. Network analysis revealed a frequent cooccurrence of *Stenotrophomonas maltophilia* with *Pseudomonas putida* and *Pseudomonas aeruginosa*. In conclusion, pangolins can be colonized with human-related ESBL-producing *K. pneumoniae* and *E. coli*. Unlike in other African wildlife, *S. aureus*-related complex was not detected in pangolins.

**IMPORTANCE** There is an ongoing debate if pangolins are a relevant reservoir for viruses such as SARS-CoV-2. Here, we wanted to know if African pangolins are colonized with bacteria that are relevant for human health. A wildlife reservoir of antimicrobial resistance would be of medical relevance in regions were consumption of so-called bushmeat is common. In 89 pangolins, we found three ESBL-producing *Klebsiella pneumoniae* strains and one ESBL-producing *Escherichia coli* strains, which were closely related to isolates from humans in Africa. This points toward either a transmission between pangolins and humans or a common source from which both humans and pangolins became colonized.

**KEYWORDS** Africa, ESBL, pangolin, antimicrobial resistance, microbiome

In the early days of the COVID-19 pandemic, pangolins were considered one of the potential animal reservoirs of SARS-CoV-2, as viral genomes with a nucleotide identity ranging from 85.5 to 92.4% were detected in *Manis javanica*, the Malayan pangolin (1). Other investigated reservoirs were bats that carried viral genomes with a much higher nucleotide identity with SARS-CoV-2 detected in humans (93 to 99.6%) (2). In addition to viruses, bats can also

Address correspondence to Frieder Schaumburg, frieder.schaumburg@ukmuenster.de.

The authors declare no conflict of interest.

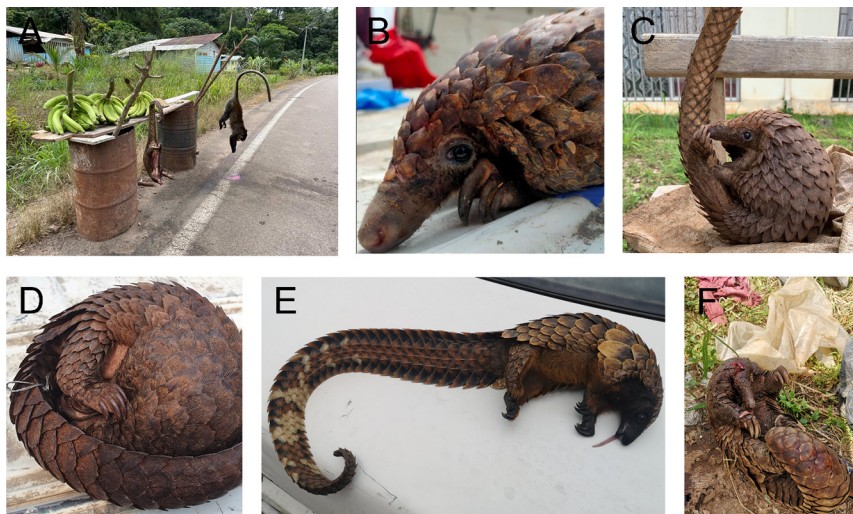

**FIG 1** Pangolins of Gabon. Pangolins were sampled on the road either dead or alive. (A to F) They belonged to *Phataginus tricuspis* (A to D), *Phataginus tetradactyla* (E), and *Smutsia gigantea* (F).

be a reservoir for bacteria. For instance, African bats can carry extended-spectrum beta-lactamase-producing *Enterobacterales* (ESBL-E) or *Staphylococcus aureus*-related complex, but it remains unknown if transmission to humans occurs (3, 4). Various bat species in Gabon (e.g., *Epomops franqueti*, *Megaloglossus woermanni*) can be colonized with ESBL-producing *Escherichia coli* or *Klebsiella pneumoniae* conferred by $bla_{CTX-M-15}$ (3). This ESBL determinant is also frequently found in humans in these areas (5). However, other wildlife (e.g., great apes, nonhuman primates) were not colonized with ESBL-E in four Gabonese national parks (6).

A new member of the *S. aureus*-related complex, *Staphylococcus schweitzeri*, was recently detected in various wildlife (e.g., bats and monkeys in Côte d'Ivoire, Democratic Republic of the Congo, Gabon, and Nigeria), raising the question of whether these animals can be a source for colonization or even infection in humans (7). So far, no infection of *S. schweitzeri* has been reported in humans (8). From a public health perspective, there is reasonable interest in having a clearer picture of bacterial colonization in African wildlife. Pangolins, for instance, are sold as a delicacy, and the scales are used for medical purposes. By touching or consuming pangolins or further processing parts of them, there is a (theoretical) risk of pathogen transmission between pangolins and humans.

To asses if pangolins could be a source for SARS-CoV-2 infections in humans on the African continent, different pangolin species sold as bushmeat were screened for corona virus variants in Gabon (2021 to 2022). Ancillary to this project, we investigated pharyngeal bacterial colonization. We consider pharyngeal colonization to be the best surrogate for the microbiome that could be a relevant source for a transmission of individual species to humans (9, 10). In contrast, bacteria from the body surface of wildlife are most likely environmental pathogens. Although the intestinal flora of animals can be diverse and rich, humans are not likely exposed to it while handling bushmeat.

Therefore, the objectives of this study were to assess the prevalence of antimicrobial-resistant species (e.g., ESBL-E) and *S. aureus*-related complex in the pharynx and to describe the bacterial community in wild Gabonese pangolins.

## RESULTS

The target number of 100 pangolins was not reached. Instead, 89 animals were included in the study between September 2021 and June 2022. The majority of the pangolins belonged to *Phataginus tricuspis* (white-bellied pangolin, *n* = 87) followed by *Phataginus tetradactyla* (black-bellied pangolin, *n* = 1) and *Smutsia gigantea* (fossorial giant pangolin, *n* = 1). In total, 47% of the pangolins (*n* = 42) were found alive, and 53% (*n* = 47) were dead (Fig. 1). Of all sampled pangolins, 51% (*n* = 45) were females, 44% (*n* = 39) were categorized as "young" or "juvenile," and 56% (*n* = 50) were adults. The median weight of the white-bellied pangolins

was 1.5 kg (range, 0.5 to 2.5 kg). The black-bellied pangolin had a weight of 2 kg, and the giant pangolin weighed 24 kg. The median length for white-bellied pangolins was 79.5 cm (range, 52 to 97.5 cm). The black-bellied pangolin was 75 cm long, and the giant pangolin was 142 cm. Samples were taken in the provinces Moyen-Ogooué ($n = 71$), Ngounié ($n = 12$), and Estuaire ($n = 6$) in Gabon.

A total of 439 bacterial isolates were included in the final analysis (see File S1 in the supplemental material for the list of species). The majority of species belonged to the genus *Pseudomonas* ($n = 170$), followed by *Stenotrophomonas* ($n = 113$) and *Achromobacter* ($n = 37$). In general, antimicrobial susceptibility rates for *Achromobacter* spp., *Pseudomonas* spp., and *Stenotrophomonas* spp. were high (Table 1). The resistance rate in *Pseudomonas* spp. against piperacillin was 15% ($n = 25/167$). Resistance rates against piperacillin in *Pseudomonas putida* isolates were up to 28% ($n = 13/47$). *Pseudomonas aeruginosa* ($n = 35$) isolates showed no resistance against the tested antibiotics. Only 2.7% ($n = 3/113$) of *Stenotrophomonas* sp. isolates were resistant against trimethoprim-sulfamethoxazole. All *Stenotrophomonas* sp. isolates were susceptible to minocycline (100%, $n = 113/113$) and levofloxacin (100%, $n = 113/113$). All analyzed *Achromobacter* sp. isolates were susceptible to piperacillin (100%, $n = 37/37$) and meropenem (100%, $n = 37/37$).

All detected *E. coli* ($n = 1$) and *K. pneumoniae* ($n = 3$) were ESBL producers. All ESBL producers were detected at two sampling sites (S00°34.496′, E010°13.334′ and S00°31.164′, E010°14.073′) located at a distance of ca. 7 km. Two of four animals were dead at the time of sampling, and all were females (File S1). The ESBL-E harbored either $bla_{CTX-M-1-group}$ (*K. pneumoniae*) or $bla_{CTX-M-9-group}$ (*E. coli*). We were only able to culture one ESBL-producing *K. pneumoniae* isolate (60007163) and one ESBL-producing *E. coli* isolate (60007166) for whole-genome sequencing (WGS). For comparative analyses to determine whether ESBL-E from pangolins cluster with isolates from humans, we randomly selected ESBL-producing *K. pneumoniae* ($n = 9$) and ESBL-producing *E. coli* ($n = 9$) from an ESBL-carrier study that was performed in the region of Lambaréné in 2010 to 2011 (5). MLST of the 10 *K. pneumoniae* isolates—in accordance with the public MLST scheme (11)—resulted in six different STs; as the pangolin isolate was ST1788, we additionally included four WGS data sets (three from Nigeria, one from the United Kingdom) of ST1788 *K. pneumoniae* published on Pathogenwatch (https://pathogen.watch/). Using the public *K. pneumoniae* core-genome multilocus sequence typing (cgMLST) scheme (see https://www.cgmlst.org), the pangolin isolate did not cluster with any human isolate from Gabon but did cluster with all ST1788 isolates from Nigeria and the United Kingdom with a minimum distance of 27 differing alleles to the most related isolate (Fig. 2A). In the remaining isolates, cgMLST-based clustering corroborated the grouping by classical MLST.

MLST of the 10 *E. coli* isolates resulted in eight different STs; of these, the ESBL-producing *E. coli* from a pangolin (ST38) was detected in the same branch as an isolate from a human carrier (ST38, Fig. 2B) (12). However, both carried a different $bla_{CTX-M}$ (isolate 60007166: $bla_{CTX-M14}$, which was chromosomally encoded; isolate 902: $bla_{CTX-M15}$, which was located on a 74-kb plasmid), and the cgMLST allelic distance between the two isolates was 414 different alleles (excluding a closer relationship) using the Enterobase *E. coli* cgMLST scheme (12). In the remaining isolates, cgMLST-based clustering corroborated the grouping by classical MLST.

No isolate belonging to the *S. aureus*-related complex was detected.

Network analysis was performed to identify frequently cooccurring species (Fig. 3). The dominant species, *Stenotrophomonas maltophilia*, commonly cooccurs with *P. putida* (32 cases; odds ratio [OR], 0.76; 95% confidence interval [CI], 0.27 to 2.09; $P = 0.64$) and *P. aeruginosa* (23 cases; OR = 0.67, 95%CI: 0.24–1.89, $P = 0.48$). Additionally, 18 cases with co-occurring *P. aeruginosa* and *P. putida* could be observed (OR = 0.91, 95%CI: 0.36–2.33, $P = 1$). However, cooccurrence of all three species could be observed in only 10 cases .

## DISCUSSION

The main findings of our study were the rare colonization with ESBL-E and the absence of *S. aureus*-related complex in the pharyngeal flora of pangolins. Instead, a frequent cooccurrence of *S. maltophilia*, *P. putida*, and *P. aeruginosa* was observed.

**TABLE 1** Antimicrobial susceptibility of bacteria from Gabonese pangolins (2021 to 2022)[a]

| Species (n) | Susceptibility [% (n)] to: | | | | | | | | | | |
|---|---|---|---|---|---|---|---|---|---|---|---|
| | Piperacillin | Ceftazidime | Ceftolozane | Imipenem | Meropenem | Ciprofloxacin | Levofloxacin | Minocycline | Tobramycin | Trimethoprim/ sulfamethoxazole | Colistin |
| Achromobacter spp. (37) | 100 (37/37) | | | | 100 (37/37) | | | | | | |
| Pseudomonas spp. (170) | 85.0 (142/167) | 94.7 (160/169) | 98.6 (144/146) | 98.7 (157/159) | 97.6 (165/169) | 100 (170/170) | | | 98.8 (167/169) | | 100 (161/161) |
| Pseudomonas aeruginosa (35) | 100 (35/35) | 100 (35/35) | 100 (35/35) | 100 (35/35) | 100 (35/35) | 100 (35/35) | | | 100 (35/35) | | 100 (35/35) |
| Pseudomonas putida (47) | 72 (33/46) | 98 (46/47) | 100 (42/42) | 96 (44/46) | 100 (47/47) | 100 (47/47) | | | 98 (46/47) | | 100 (46/46) |
| Stenotrophomonas spp. (113) | | | | | | | 100 (113/113) | 100 (113/113) | | 97.3 (110/113) | |
| Stenotrophomonas maltophilia (63) | | | | | | | 100 (63/63) | 100 (63/63) | | 98 (62/63) | |

[a]Species/genera were only considered if ≥20 isolates were available in each group. For some species, AST with Vitek2 was not possible for all antibiotics (e.g., absence of EUCAST breakpoints). *P. aeruginosa*, *P. putida*, and *S. maltophilia* are represented twice in the table: once as the individual species and once as a member of the respective genus.

# A. *Klebsiella pneumoniae*

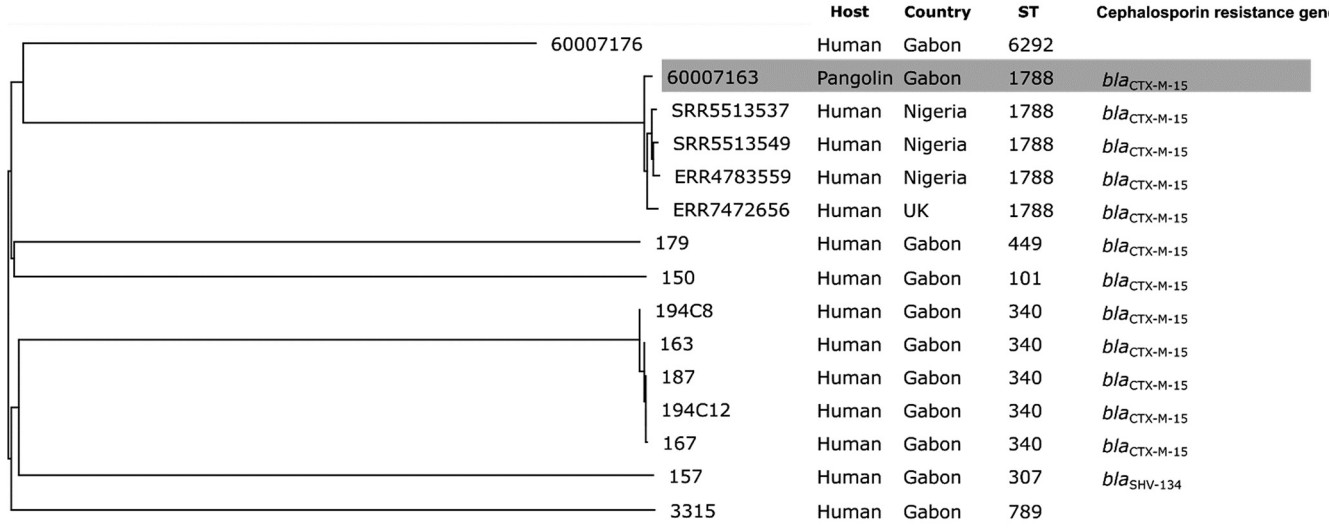

| | Host | Country | ST | Cephalosporin resistance gene |
|---|---|---|---|---|
| 60007176 | Human | Gabon | 6292 | |
| 60007163 | Pangolin | Gabon | 1788 | $bla_{CTX-M-15}$ |
| SRR5513537 | Human | Nigeria | 1788 | $bla_{CTX-M-15}$ |
| SRR5513549 | Human | Nigeria | 1788 | $bla_{CTX-M-15}$ |
| ERR4783559 | Human | Nigeria | 1788 | $bla_{CTX-M-15}$ |
| ERR7472656 | Human | UK | 1788 | $bla_{CTX-M-15}$ |
| 179 | Human | Gabon | 449 | $bla_{CTX-M-15}$ |
| 150 | Human | Gabon | 101 | $bla_{CTX-M-15}$ |
| 194C8 | Human | Gabon | 340 | $bla_{CTX-M-15}$ |
| 163 | Human | Gabon | 340 | $bla_{CTX-M-15}$ |
| 187 | Human | Gabon | 340 | $bla_{CTX-M-15}$ |
| 194C12 | Human | Gabon | 340 | $bla_{CTX-M-15}$ |
| 167 | Human | Gabon | 340 | $bla_{CTX-M-15}$ |
| 157 | Human | Gabon | 307 | $bla_{SHV-134}$ |
| 3315 | Human | Gabon | 789 | |

⊢————⊣ 100 alleles distance

# B. *Escherichia coli*

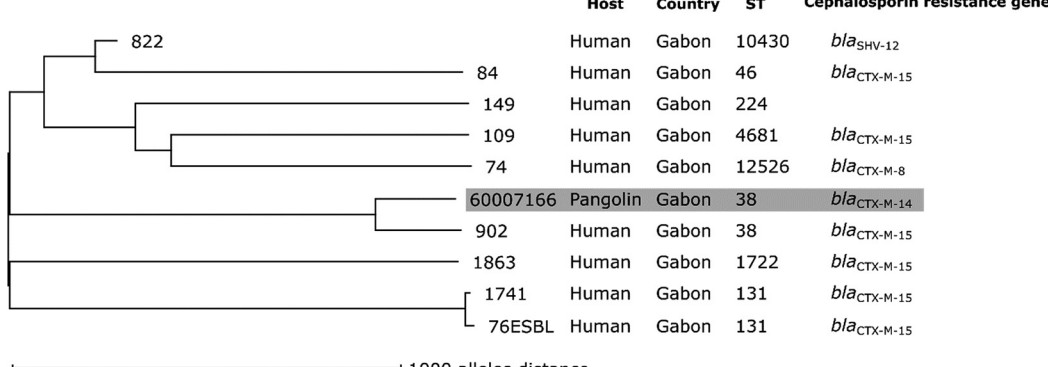

| | Host | Country | ST | Cephalosporin resistance gene |
|---|---|---|---|---|
| 822 | Human | Gabon | 10430 | $bla_{SHV-12}$ |
| 84 | Human | Gabon | 46 | $bla_{CTX-M-15}$ |
| 149 | Human | Gabon | 224 | |
| 109 | Human | Gabon | 4681 | $bla_{CTX-M-15}$ |
| 74 | Human | Gabon | 12526 | $bla_{CTX-M-8}$ |
| 60007166 | Pangolin | Gabon | 38 | $bla_{CTX-M-14}$ |
| 902 | Human | Gabon | 38 | $bla_{CTX-M-15}$ |
| 1863 | Human | Gabon | 1722 | $bla_{CTX-M-15}$ |
| 1741 | Human | Gabon | 131 | $bla_{CTX-M-15}$ |
| 76ESBL | Human | Gabon | 131 | $bla_{CTX-M-15}$ |

⊢————————————⊣ 1000 alleles distance

**FIG 2** (A and B) Neighbor-joining trees of ESBL-producing *K. pneumoniae* (A) and *E. coli* (B) from pangolins and humans in Gabon based on allelic profiles of the respective cgMLST scheme. Nodes were labeled with isolate designation, host, country of origin, MLST ST, and if applicable, with the cephalosporin-resistance genes. Missing cgMLST targets were ignored in pairwise comparisons. Isolates from pangolins are highlighted in gray.

Although ESBL-producing *K. pneumoniae* ST1788 ($bla_{CTX-M\ 15}$) did not cluster with isolates from humans in Gabon, this ST was reported three times from human blood cultures in Nigeria and one from a urine sample in the United Kingdom (Fig. 2). In addition, one ESBL-producing ST1788 *K. pneumoniae* isolate was also isolated from a urinary tract infection in Ghana (13). ESBL-producing ST38 *E. coli* is a known pandemic clone and is mainly reported from extraintestinal infections in Africa (14, 15). Our findings suggests that the ESBL-producing *K. pneumoniae* and ESBL-producing *E. coli* from pangolins might share a similar background with isolates from humans. If they evolved separately or if they were transmitted from animals to humans (or vice versa) remains unclear. Transmission of ESBL-E from humans to pangolins and further spread among other wildlife species is, however, unlikely, as pangolins are usually hunted for food, and release after capture would only happen accidentally.

Data on the normal flora of pangolins are scarce, but some reports, particularly from Asian pangolins, suggest that *E. coli*, *K. pneumoniae*, and *Pseudomonas* spp. are common commensals (16). Therefore, it is likely that pangolins can be a stable reservoir also for antimicrobial-resistant variants of *E. coli* and *K. pneumoniae* (e.g., ESBL producers). Few studies have addressed the intestinal microbiome of pangolins using metagenomics approaches. These studies revealed that the predominant species of the pharyngeal flora (*S. maltophilia*,

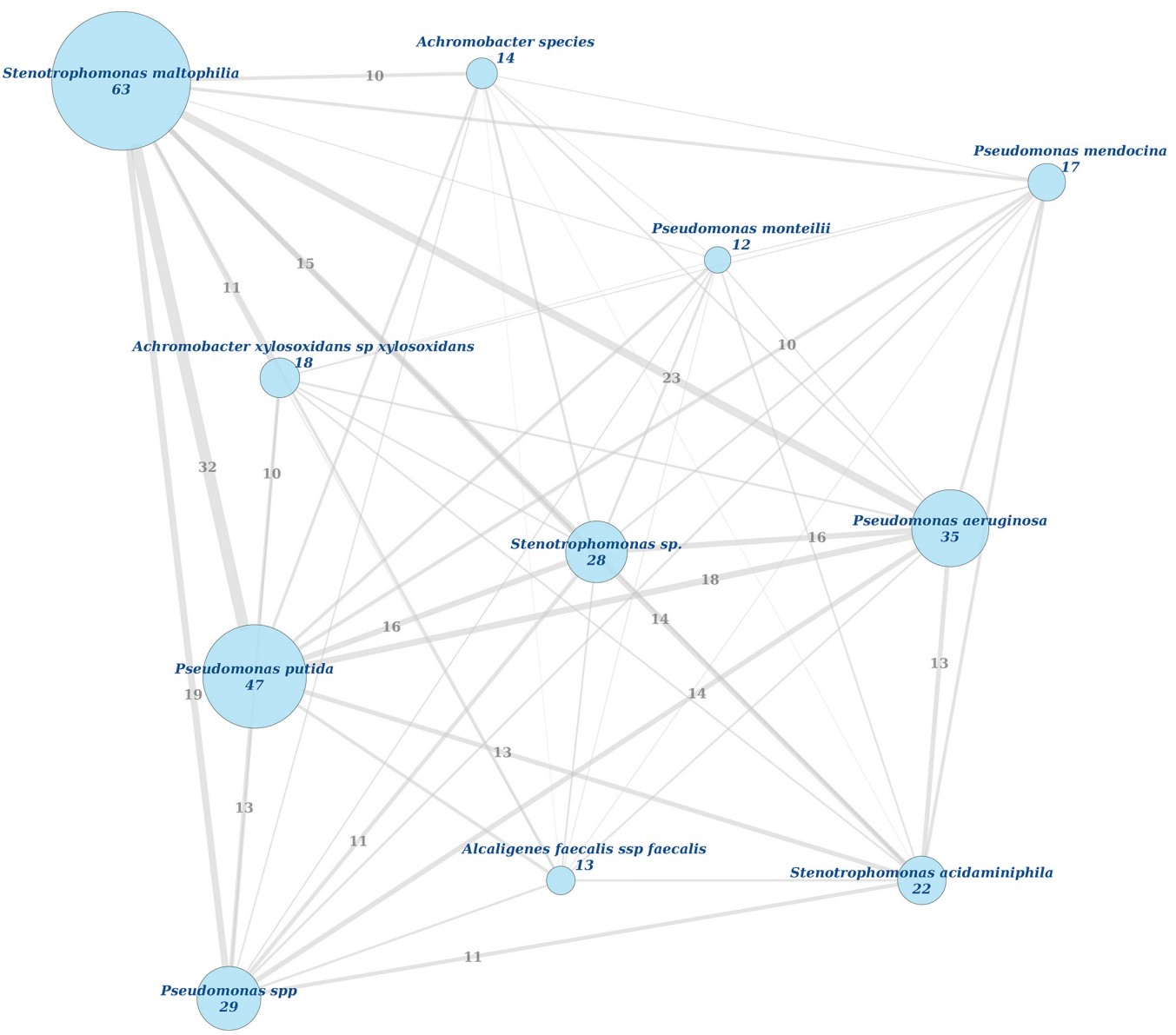

**FIG 3** Network analysis of bacteria detected in pangolins in Gabon. Only species with ≥10 isolates were included. Node size and edge thickness are correlated with the number of observations. Blue numbers report the total number of isolates of the species; gray numbers report the number of cooccurrences.

*P. putida*, and *P. aeruginosa*) are rarely detected in the intestinal microbiome, where *Firmicutes*, *Bacteroidetes*, and *Proteobacteria* are more common (17).

Our study has limitations: First, we might have underestimated the prevalence of wild-type isolates (e.g., *E. coli*) since we used selective agar plates. This procedure was chosen due to the overgrowth with environmental species, which impaired the detection of more fastidious species. Second, the comparison of ESBL-E from pangolins and humans from Gabon should be interpreted with caution due to the time difference of approximately 10 years. In addition, we were only able to perform WGS on a very small number of isolates, which further limits general conclusions. Third, the samples had to be sent to Germany and could not be examined on site, which can lead to quality losses due to the longer time span from sampling to analysis. Fourth, nonculturable bacteria were not taken into account.

**Conclusion.** In conclusion, pangolins can be colonized with human-related ESBL-producing *K. pneumoniae* and *E. coli*. We did not find any evidence that pangolins are a reservoir for the *S. aureus*-related complex, unlike other African wildlife.

## MATERIALS AND METHODS

**Approval.** The approval to take samples from pangolins was issued by the Ministère de l'Enseignement Supérieur et de la Recherche Scientifique, du Transfert des Technologies, de l'Education Nationale, Chargé de la Formation Civique (no. AR025/21/MESRSTTENCFC/CENAREST/CG/CST/CSAR).

**Sample size calculation.** In the absence of any pilot data, a specific sample size calculation was not done. A convenience sample of 100 animals was deemed to be appropriate for the objectives.

**Animals.** Animals were included in this study based on availability and the acceptance of the hunter or the seller to sample the pangolins. Swabs were taken where the animals were found or at the Centre de Recherches Médicales de Lambaréné. Living animals were bought from sellers and were set free after sampling.

For each animal, the following data were recorded: date of sampling, the GPS location of the geographical sampling site using the mobile application GPS Essentials, state of living (dead/alive), sex (female/male), age (young/adult), weight, and length.

Pharyngeal swabs were taken with sterile cotton tips and stored in Amies transport medium (Transwab, Check Diagnostics, Bad Oldesloe, Germany). Samples were stored at 4°C for a maximum of 4 to 6 weeks until further analyses.

**Culture and susceptibility testing.** Swabs were sent to Germany within 5 days at room temperature for culture and susceptibility testing. In a pilot study, we used a set of selective and nonselective agar plates, i.e., Columbia blood agar, chocolate agar, and McConkey agar (all BD, Heidelberg, Germany), with and without enrichment in thioglycolate broth (BD). Since these culture plates were overgrown with primarily environmental species (e.g., *Myroides, Alcaligenes*) we decided to use more selective media. Any phenotypically different colonies growing on chromID ESBL, SAIDE (bioMérieux, Marcy l'Étoile, France), colistin-nalidixin agar (BD), and cetrimide agar (Oxoid) were subcultured for species identification using matrix-assisted laser desorption ionization–time of flight (MALDI-TOF) mass spectrometry (Biotyper Sirius One, Bruker, Bremen, Germany) and the MBT-Compass IVD database (4.2). Isolates with an identifications score of >2.00 were included in the study without further analyses. Isolates with an identification score of <1.7 were retested. Scores between 1.7 and 2 were only used if the identification passed the internal plausibility assessment (e.g., phenotypic appearance, expected resistance/susceptibility phenotypes, etc.).

Antimicrobial susceptibility testing (AST) was done with a Vitek2 automated system (bioMérieux) and antimicrobial susceptibility testing (AST) test cards N389 (*Acinetobacter* spp., *Alcaligenes* spp., *Pseudomonas* spp.) or N214 (*Enterobacterales*). Disk diffusion was done for *Stenotrophomonas* spp. and *Achromobacter* spp. using EUCAST clinical breakpoints for *Stenotrophomonas maltophilia* and *Achromobacter xylosoxidans*, respectively. If automated AST testing was unsuccessful, the isolate was subjected to disk diffusion. AST was done according to EUCAST guidelines, and test results were interpreted using EUCAST clinical breakpoints (version 11.0 and 12.0 [18]). Antimicrobial agents that were tested as "I" (susceptible increased exposure according to EUCAST) were considered susceptible when calculating susceptibility rates.

Only levofloxacin and minocycline were tested and interpreted according to CLSI for *Stenotrophomonas* spp. (19).

*Enterobacterales* resistant to 3rd-generation cephalosporins were further screened for ESBL determinants using eazyplex SuperBug CRE (Amplex, Gahrs-Bahnhof, Germany).

Only species or genera with ≥20 isolates were entered in the calculation of susceptibility rates (Table 1) (20).

**Whole-genome sequencing.** Only ESBL-E were subjected to whole-genome sequencing (WGS) as described recently (21). Subsequently, we performed multilocus sequence typing (MLST) and core genome MLST (cgMLST) if a species-specific scheme was available at https://www.cgmlst.org (22). Based on the allelic profiles of the cgMLST scheme, we constructed a neighbor-joining tree to display the clonal relationship among the ESBL-E and, if applicable, additional reference strains using Ridom SeqSphere+ software version 7 (Ridom GmbH, Münster, Germany) (23). For extraction of genomic determinants related to the ESBL phenotype, we used the AMRFinder (24) that is implemented in SeqSphere+. The genome sequences determined here were available at https://www.cgmlst.org.

**Network analysis.** Network analysis was performed using R version 4.2.2 and the R package igraph (25, 26). Adapting the visualization of cooccurring and persistent symptoms in COVID-19 (27), interaction networks were generated: every species is represented by a node; cooccurring species within one animal are represented by an edge. Node size and edge thickness were correlated with the respective number of observations. Precise numbers are additionally reported (species, all; intersects, ≥10). A missing edge between two nodes indicates that cooccurrence of the two species was never observed in any animal. To reduce the complexity of the network and improve visualization, only species detected in ≥10 animals were considered. Cooccurrence was further analyzed, determining odds ratios, 95% confidence intervals (CIs) and $P$ values using the R package questionr (28). Observation of species 1 (yes/no) versus species 2 (yes/no) in the total population of $n = 89$ was considered.

**Data availability.** The data used for this analysis are presented in the manuscript and File S1. cgMLST data are available at https://www.cgmlst.org.

## SUPPLEMENTAL MATERIAL

Supplemental material is available online only.

**SUPPLEMENTAL FILE 1**, XLSX file, 0.1 MB.

## ACKNOWLEDGMENTS

We thank the laboratory technicians of the Institute of Medical Microbiology and Hygiene for their excellent work and support.

Johanna P. Wiethoff: investigation, data curation, writing–original draft, writing–review and editing. Sarah Sandmann: validation, methodology, visualization, writing–original draft, writing–review and editing. Tom Theiler: data curation, writing–original draft, writing–review and editing. Chimene Nze Nkogue: resources, investigation, writing–review and editing. Etienne-François Akomo-Okouef: resources, investigation, writing–review and editing. Julian Varghese: validation, methodology, writing–original draft. Andrea Kreidenweiss: conceptualization, resources, writing–review and editing. Alexander Mellmann: validation, resources, supervision, methodology, writing–original draft, writing–review and editing. Bertrand Lell: resources, supervision, writing–review and editing. Ayola A. Adegnika: resources, supervision, writing–review and editing. Jana Held: conceptualization, resources, supervision, writing–original draft, writing–review and editing. Frieder Schaumburg: conceptualization, resources methodology, visualization, formal analysis, investigation, writing–original draft, writing–review and editing.

This study was supported by institutional funds. We acknowledge the open access fund of the University of Münster.

We declare no competing interest.

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
