## [Reviewer comments · Microbiology Spectrum]

Microbiology Spectrum

Pharyngeal communities and antimicrobial resistances in Pangolins, Gabon

Johanna Wiethoff, Sarah Sandmann, Tom Theiler, Chimene Nkogue, Etienne-Francois Akomo-Okoue, Julian Varghese, Andrea Kreidenweiss, Alexander Mellmann, Bertrand Lell, Ayôla Adegnika, Jana Held, and Frieder Schaumburg

Corresponding Author(s): Frieder Schaumburg, Westfälische Wilhelms-Universität Münster

Review Timeline:

Submission Date:	February 13, 2023
Editorial Decision:	April 26, 2023
Revision Received:	May 22, 2023
Accepted:	June 5, 2023

Editor: Katharina Schaufler

Reviewer(s): The reviewers have opted to remain anonymous.

Transaction Report:

DOI: <https://doi.org/10.1128/spectrum.00664-23>

April 26, 2023

Dr. Frieder Schaumburg
Westfälische Wilhelms-Universität Münster
Institute for Medical Microbiology
Domagkstr. 10
Münster, North-Rhine Westphalia 48149
Germany

Re: Spectrum00664-23 (Microbial communities and antimicrobial resistances in Pangolins, Gabon)

Dear Dr. Frieder Schaumburg:

Link Not Available

Sincerely,

Katharina Schaufler

Journals Department
Reviewer comments:

Reviewer #2 (Public repository details (Required)):

It would be beneficial if the authors reported all the species/genus identified in the study. The exact number of isolates identified by MALDI-TOF needs to be clarified by the authors.

Reviewer #2 (Comments for the Author):

The manuscript is well written and provides sufficient background information. It reports the levels of antimicrobial resistance (AMR) in bacteria isolated from the pharynx of pangolins in Gabon. The authors report low levels of AMR in the 439 isolates

investigated with 3 ESBL-producing *K. pneumoniae* and 1 ESBL-producing *E. coli* being described. Having investigated 89 animals and 439 isolates, the sample size is one big strength of this study. However more emphasis could have been given to the species found in the samples investigated, as studies on bacterial communities in pangolins' pharynx are lacking. Additionally, the discussion could benefit from further discussing public health risks (including AMR dissemination) associated with dealing with wild pangolins, in my perspective.

Please see my comments below:

a) Generic comments

- Title: Please consider rephrasing the title to better reflect the study conducted (bacteria in pharyngeal swabs and selective media)
- Importance: It may be relevant to mention the role of wildlife as potential reservoirs of resistant bacteria and give more emphasis to AMR.
- Throughout the manuscript the authors use "sp." Can the authors confirm if they mean species as plural "spp.", please?

b) Specific comments

Abstract:

Line 32: Can the author use "Gram" instead of "gram", please?

Introduction:

- Can the authors add more information on the importance of pharyngeal bacteria and how these bacteria can/may spread to humans, please?
- Can the authors give more emphasis to the AMR issue, please? The study design of this study has been tailored by the SARS-CoV-2 study however the authors can focus more on AMR.

Lines 59 and 60: Please consider using "(...) viral genomes with a nucleotide identity ranging from 85.5% to 92.4% were detected (...)"

Line 64: Can the authors use "extended" instead of "Extended", please?

Line 65: Can the authors replace "unresolved" with "unknown", please?

Line 76: Can the authors use "public health", please?

Lines 75-77: The authors mention "Therefore, from a public-health perspective, there is reasonable interest in having a clearer picture of bacterial colonization in African wildlife". Can the authors add a sentence explaining the relationship between human population and pangolins/wildlife, please?

Line 80: Can the authors consider using "project" instead of "investigation", please?

Line 82: Can the authors remove "ancillary", please?

Line 83: Can the authors please add "(...) and *S. aureus*-related complex in the pharynx of pangolins, and to (...)"?

Methods:

Line 87: Can the authors confirm if this refers to ethical approval, please?

Line 97: Can the authors add "(...) were included in this study (...)", please?

Line 105: Can the authors clarify if the cotton tips were sterile, please?

Line 107: Can the authors clarify for how long these samples were stored at 4C, please?

Lines 110 and 111: Can the authors please confirm if the samples were sent at room temperature, please? Additionally, how long did it take on average?

Lines 113 to 118: Can the authors confirm if only selective media chromID ESBL, ceftrimide agar and colistin-nalidixin agar were used, please?

Line 117: Can the authors use "ceftrimide", please?

Line 118: Can the authors confirm how many isolates were identified by MALDI-TOF, please? Can the authors also add some information on the quality of the identification, please?

Line 119: Can the authors confirm if 4.2 is the version, please?

Lines 122 to 131: Can the authors confirm if they took into consideration the intrinsic resistance of some species when performing the AST, please?

Lines 122 to 134: Can the authors add references, please?

Line 139: Can the authors add the number (n=), please?

Line 142 and 145: Can the authors add a reference, please?

Line 152: Can the authors expand/justify this choice, please?

Results:

Line 164: Can the authors clarify why 89 out of 100 pangolins were included in this study, please?

Line 170: Can the authors use ""adult"" or "adults", please?

Line 176: The authors mention that 439 isolates were included. Out of how many? It could be useful to provide a list of all species/genus isolated in this study (potentially as supplemental material).

Lines 178 to 180: Can the authors clarify if they mean antimicrobial susceptibility rates or resistance rates, please?

Line 181 to 182: Can the authors consider " Resistance rate against piperacillin in *Pseudomonas putida* isolates were up to 28%"

Line 189: Can the authors clarify how many E. coli and Klebsiella isolates were found in this study, please? Can the authors also clarify if the selection of the 4 isolates for WGS was based on the ESBL phenotype provided by Vitek2, please?

Lines 198 and 202: Can the authors add "the" before "UK", please?

Lines 209 to 211: Can the authors consider using "and the cgMLST allelic distance between the two isolates was 414 different alleles (excluding a closer relationship) using the Enterobase E. coli cgMLST scheme", please?

Discussion:

Line 225: Can the authors remove "is", please?

Line 239 to 244: Can the authors please consider other limitations including the number of isolates subjected to WGS, testing limitations associated with using selective media, storage/transport, non-culturable bacteria etc.?

Additionally, can the authors discuss the public health risks associated with dealing with wild pangolins. Can pangolins contact with humans and get AMR isolates from humans due to management practices, food habits etc.? Isolates were mainly susceptible which is an important finding that can be further discussed.

Tables:

Line 355: Can the authors please confirm if the issues with table 1 are due to formatting at editing stage? Can the authors provide an improved version of the table, please?

Line 356: The authors mention that 439 isolates were investigated in this study however in Table 1, 465 isolates are referred. Can the authors clarify this, please?

Lines 356 to 357: Can the authors clarify the following, please? The authors mention that for some species, it was not possible to perform AST with VITEK2. Was this because of limitations of the VITEK cards used and was disk diffusion performed? Or was this an issue due to the lack of breakpoints in the literature for certain species?

Line 366: Can the authors use "cephalosporin-resistance", please?

Reviewer #3 (Comments for the Author):

Importance and Introduction:

1. Care should be taken to differentiate this work from the SARS-CoV-2 work performed on these same specimen. Too often the SARS-CoV-2 work was mentioned unnecessarily.

2. Pg4 Ln72: discussion of new S. aureus complex, but unclear if it is relevant to human health

3. Introduction seemed rushed and emphasized wrong topics. Rationale for bacterial and AMR work weak.

4. More discussion about pangolin microbiome would be helpful

Methods:

1. Sample size calculation: why 100? Why not 50? Or 80? Does not meeting the 100 specimen target lessen the results?

Results:

1. Initial results confusing as presented. Would be beneficial to provide total list of organisms identified by MALDI-TOF prior to discussion of AMR.

2. Result would benefit from analysis of trends for AMR: more AMR/species found in dead animals? Live animals? Region?

3. Were all strains identified by MALDI-TOF and then sequenced? Unclear as presented.

4. Why were the specific antibiotics chosen for each genus?

5. The networking analysis for co-occurrence is very interesting.

6. Overall, Results could be better organized to more effectively show process and information

Discussion

1. Discussion is rushed and should be better organized; discussion on totality of data, not just Klebsiella and E. coli.

Staff Comments:

Preparing Revision Guidelines

Please return the manuscript within 60 days; if you cannot complete the modification within this time period, please contact me. If you do not wish to modify the manuscript and prefer to submit it to another journal, please notify me of your decision immediately so that the manuscript may be formally withdrawn from consideration by Microbiology Spectrum.

Universitätsklinikum Münster . 48129 Münster . [42800]

To
Prof. Katharina Schaufler
Editor, Microbiology Spectrum
Journals Department
Universitätsklinikum Münster
Institut für Medizinische Mikrobiologie
Univ.-Prof. Dr. med. Frieder Schaumburg
Direktor

Domagkstraße 10
48149 Münster
www.ukm-lageplan.de

T +49 251 83-52767
Servicezentrale: T +49 251 83-55555

frieder.schaumburg@ukmuenster.de
www.ukm.de

Spectrum00664-23

Münster, 05.06.2023

Dear Editor,

we thank you and the two reviewers for thoroughly reviewing our manuscript. We appreciate the constructive suggestions and fair evaluation.

In the following paragraphs, we reply point-by-point to the reviewers suggestions.

Reviewer #2

It would be beneficial if the authors reported all the species/genus identified in the study. The exact number of isolates identified by MALDI-TOF needs to be clarified by the authors.

Reply: We now upload the database with all species and corresponding antimicrobial test results.

The manuscript is well written and provides sufficient background information. It reports the levels of antimicrobial resistance (AMR) in bacteria isolated from the pharynx of pangolins in Gabon. The authors report low levels of AMR in the 439 isolates investigated with 3 ESBL-producing *K. pneumoniae* and 1 ESBL-producing *E. coli* being described. Having investigated 89 animals and 439 isolates, the sample size is one big strength of this study. However, more emphasis could have been given to the species found in the samples investigated, as studies on bacterial communities in pangolins' pharynx are lacking. Additionally, the discussion could benefit from further discussing

public health risks (including AMR dissemination) associated with dealing with wild pangolins, in my perspective.

Reply: Thank you for your kind evaluation. We will address the issues in detail below.

a) Generic comments

- Title: Please consider rephrasing the title to better reflect the study conducted (bacteria in pharyngeal swabs and selective media)

Reply: We changed the title better reflect the fact that we looked at the pharyngeal flora. However, we feel that the title would be too extended if we also address the use of the media.

- Importance: It may be relevant to mention the role of wildlife as potential reservoirs of resistant bacteria and give more emphasis to AMR.

Reply: We now added a sentence on the importance of AMR in wildlife particularly in regions where the consumption of bushmeat is common.

- Throughout the manuscript the authors use "sp." Can the authors confirm if they mean species as plural "spp.", please?

Reply: Thank you for pointing out this issue. We now corrected the manuscript accordingly.

b) Specific comments

Abstract:

Line 32: Can the author use "Gram" instead of "gram", please?

Reply: Corrected as suggested.

Introduction:

- Can the authors add more information on the importance of pharyngeal bacteria and how these bacteria can/may spread to humans, please?

Reply: We now provide an explanation why we consider the pharyngeal flora as the most relevant flora for a transmission of bacterial species from wildlife to humans. This is, also now supported by two additional references.

- Can the authors give more emphasis to the AMR issue, please? The study design of this study has been tailored by the SARS-CoV-2 study however the authors can focus more on AMR.

Reply: As indicated in the introduction, we did not only focus on AMR but also on other bacteria (even antimicrobial susceptible isolates) that could be of relevance for humans. We therefore want to keep the introduction as broad as possible without putting the focus too much on AMR. We hope, that the reviewer agrees.

Lines 59 and 60: Please consider using "(...) viral genomes with a nucleotide identity ranging from 85.5% to 92.4% were detected (...)"

Reply: The sentence was corrected as suggested.

Line 64: Can the authors use "extended" instead of "Extended", please?

Reply: Corrected as suggested.

Line 65: Can the authors replace "unresolved" with "unknown", please?

Reply: Corrected as suggested.

Line 76: Can the authors use "public health", please?

Reply: Corrected as suggested.

Lines 75-77: The authors mention "Therefore, from a public-health perspective, there is reasonable interest in having a clearer picture of bacterial colonization in African wild-life". Can the authors add a sentence explaining the relationship between human population and pangolins/wildlife, please?

Reply: We now provide an explanation about the relationship between the human population and pangolins in Gabon. Pangolins are sold as a delicacy and the scales are used for medical purposes. By touching or consuming pangolins or further processing parts of them, there is a (theoretical) risk of pathogen transmission to humans.

Line 80: Can the authors consider using "project" instead of "investigation", please?

Reply: Corrected as suggested.

Line 82: Can the authors remove "ancillary", please?

Reply: Corrected as suggested.

Line 83: Can the authors please add "(...) and *S. aureus*-related complex in the pharynx of pangolins, and to (...)?"

Reply: The sentence was corrected as suggested.

Methods:

Line 87: Can the authors confirm if this refers to ethical approval, please?

Reply: Since no human samples were included in the study, an approval from the ethical committee was not mandatory according to Gabonese law. Instead, the "Ministère de l'Enseignement supérieur et de la Recherche scientifique, du Transfert des Technologies, de l'Education nationale, chargé de la Formation Civique" is in charge to approve research on animals in Gabon.

Line 97: Can the authors add "(...) were included in this study (...)", please?

Reply: The sentence was corrected as suggested.

Line 105: Can the authors clarify if the cotton tips were sterile, please?

Reply: Yes, according to the manufacturer, the cotton tips are sterile. This is now clarified in the revised version.

Line 107: Can the authors clarify for how long these samples were stored at 4°C, please?

Reply: We now clarify the storage duration (4-6 weeks).

Lines 110 and 111: Can the authors please confirm if the samples were sent at room temperature, please? Additionally, how long did it take on average?

Reply: Yes, the samples were sent at room temperature and arrived the destination within five days. This information is now added in the manuscript.

Lines 113 to 118: Can the authors confirm if only selective media chromID ESBL, cetrimide agar and colistin-nalidixin agar were used, please?

Reply: As indicated in the manuscript, we used chromID ESBL (for ESBL-E), SAIDE (for *S. aureus* related complex), colistin-nalidixin agar (for any Gram-positive bacteria) and cetrimid agar (for Gram-negative and *Pseudomonas* spp.).

We used this set of agar as non-selective media were completely overgrown by environmental species as shown in the section on “Culture and susceptibility testing” (pilot study).

Line 117: Can the authors use "cetrimide", please?

Reply: Corrected as suggested.

Line 118: Can the authors confirm how many isolates were identified by MALDI-TOF, please?

Reply: we now upload the complete list of all isolates as a supplement.

Can the authors also add some information on the quality of the identification, please?

Reply: We now clarify this point and write in the revised version: “Only isolates with an identifications score >2.00 were included in the study. Isolates with an identification score of <1.7 were retested. Scores between 1.7 and 2 were only used if the identification passed the internal plausibility assessment (e.g. phenotypic appearance, expected resistance/susceptibility phenotypes etc.).”

Line 119: Can the authors confirm if 4.2 is the version, please?

Reply: yes, confirmed.

Lines 122 to 131: Can the authors confirm if they took into consideration the intrinsic resistance of some species when performing the AST, please?

Reply: Yes, of course, this is part of the plausibility check. See our comment above.

Lines 122 to 134: Can the authors add references, please?

Reply: We now add a references for CLSI and EUCAST breakpoints. There is no reference for “eazyplex® SuperBug CRE” as it is a commercial kit.

Line 139: Can the authors add the number (n=), please?

Reply: We tend to adhere strictly to the structure of methods and results. Here, we only report the methods. See the result section for the total numbers of detected isolates. Three ESBL-producing *K. pneumoniae* (*bla*_{CTX-M-1} group) and one ESBL-producing *E. coli* (*bla*_{CTX-M-9} group) were detected.

Line 142 and 145: Can the authors add a reference, please?

Reply: An additional reference for cgMLST and the software SeqSphere is now given.

Line 152: Can the authors expand/justify this choice, please?

Reply: This only refers to the visualization. This is just a standard way to link those items that are co-detected. In our institution, network analyses were first implemented during the COVID-pandemic and we here refer to this method. We already provide a detailed description of how the network analysis is displayed and how it is interpreted. We should be grateful if the reviewer could explain better what is missing in our description.

Results:

Line 164: Can the authors clarify why 89 out of 100 pangolins were included in this study, please?

Reply: Thank you for pointing out a misleading description. We aimed to include 100 animals but achieved only 89. This is now clarified in the revised version of the manuscript.

Line 170: Can the authors use ""adult"" or "adults", please?

Reply: Corrected as suggested.

Line 176: The authors mention that 439 isolates were included. Out of how many? It could be useful to provide a list of all species/genus isolated in this study (potentially as supplemental material).

Reply: We appreciate the suggestion and now provide the complete list of isolates as a supplementary information.

Lines 178 to 180: Can the authors clarify if they mean antimicrobial susceptibility rates or resistance rates, please?

Reply: We agree that it is confusing to report both resistance and susceptibility rates. For clarification, we now report (in line with table 1) susceptibility rates and only point out those antimicrobials that showed a relevant resistance in the tested isolates.

Line 181 to 182: Can the authors consider " Resistance rate against piperacillin in *Pseudomonas putida* isolates were up to 28%"

Reply: The sentence was corrected as suggested.

Line 189: Can the authors clarify how many *E. coli* and *Klebsiella* isolates were found in this study, please? Can the authors also clarify if the selection of the 4 isolates for WGS was based on the ESBL phenotype provided by Vitek2, please?

Reply: We now provide the total number of *E. coli* (n=1) and *K. pneumoniae* (n=3). They were confirmed to be ESBL-producers using the commercial amplex test. Only those isolates that carried an ESBL-gene were subjected to WGS.

Lines 198 and 202: Can the authors add "the" before "UK", please?

Reply: Corrected as suggested.

Lines 209 to 211: Can the authors consider using "and the cgMLST allelic distance between the two isolates was 414 different alleles (excluding a closer relationship) using the Enterobase *E. coli* cgMLST scheme", please?

Reply: The sentence was corrected as suggested.

Discussion:

Line 225: Can the authors remove "is", please?

Reply: Corrected as suggested.

Line 239 to 244: Can the authors please consider other limitations including the number of isolates subjected to WGS, testing limitations associated with using selective media, storage/transport, non-culturable bacteria etc.?

Reply: We agree with the reviewers points and now mention the suggested limitations in the revised version.

Additionally, can the authors discuss the public health risks associated with dealing with wild pangolins. Can pangolins contact with humans and get AMR isolates from humans due to management practices, food habits etc.? Isolates were mainly susceptible which is an important finding that can be further discussed.

Reply: We now address this point in the revision and write: A transmission of ESBL-producers from humans to pangolins and a further spread among other wildlife species is unlikely as pangolins are usually hunted for food, a release after capture would only happen accidentally.

Tables:

Line 355: Can the authors please confirm if the issues with table 1 are due to formatting at editing stage? Can the authors provide an improved version of the table, please?

Reply: We cannot identify any formatting problem with table 1.

Line 356: The authors mention that 439 isolates were investigated in this study however in Table 1, 465 isolates are referred. Can the authors clarify this, please?

Reply: Indeed, 439 isolates were recovered. The misleading information of table 1 is that for instance *P. aeruginosa* is calculated twice: once as the species “*P. aeruginosa*” and once as the genus “*Pseudomonas spp.*” This is now clarified in a footnote to table 1.

In addition, the table only includes those species/genus with ≥ 20 isolates.

Lines 356 to 357: Can the authors clarify the following, please? The authors mention that for some species, it was not possible to perform AST with VITEK2. Was this because of limitations of the VITEK cards used and was disk diffusion performed? Or was this an issue due to the lack of breakpoints in the literature for certain species?

Reply: AST was not possible using Vitek as there are no breakpoints/validations from the manufacturer. This is now clarified in the footnote of the table.

Line 366: Can the authors use "cephalosporin-resistance", please?

Reply: Corrected as suggested.

Reviewer #3 (Comments for the Author):

Importance and Introduction:

1. Care should be taken to differentiate this work from the SARS-CoV-2 work performed on these same specimen. Too often the SARS-CoV-2 work was mentioned unnecessarily.

Reply: We agree. However, we feel it is important to understand the circumstances why the study was done as an ancillary study. Therefore, we think that the study on SARS-CoV-2 should be mentioned here briefly.

2. Pg4 Ln72: discussion of new *S. aureus* complex, but unclear if it is relevant to human health

Reply: We agree and address this point by writing “So far, no infection of *S. schweitzeri* was reported in humans” with a reference to the work by Grossmann et al.

3. Introduction seemed rushed and emphasized wrong topics. Rationale for bacterial and AMR work weak.

Reply: We must admit that this is a matter of opinion. We think it is good and right to discuss critical points thoroughly. However, a good discussion is occasionally limited by a poor data as we have in relation to African wildlife.

4. More discussion about pangolin microbiome would be helpful

Reply: we updated our PubMed search on the microbiome of pangolins and identified only studies on the fecal microbiome. The most recent relevant study by Jiao W, Liu L, Zeng Z, Li L, Chen J. Front Microbiol. 2022 is now discussed in the revised version.

Methods:

1. Sample size calculation: why 100? Why not 50? Or 80? Does not meeting the 100 specimen target lessen the results?

Reply: As stated in the methods section, an evidence based sample size calculation was not done. The total size of n=100 is a convenience sample. We now clarify this in the revised version.

Results:

1. Initial results confusing as presented. Would be beneficial to provide total list of organisms identified by MALDI-TOF prior to discussion of AMR.

Reply: We agree, see also our comment to reviewer #2. We now provide the complete list of species as a supplementary information. This is presented prior to the discussion of AMR.

2. Result would benefit from analysis of trends for AMR: more AMR/species found in dead animals? Live animals? Region?

Reply: This is a good point. Indeed, almost all ESBL-producers were detected in animals from one site (Koungoulé, S00°31.164', E010°14.073', 291ft), half of them were dead, all were females. This is now addressed in the revised version.

3. Were all strains identified by MALDI-TOF and then sequenced? Unclear as presented.

Reply: No, only ESBL-producing Enterobacterales were subjected to WGS. This should be clearer now by the statement “Only ESBL-E were subjected to whole genome sequencing (WGS) as described recently”.

4. Why were the specific antibiotics chosen for each genus?

Reply: Yes, the selection was done according to EUCAST and CLSI. References for these two guidance documents are now provided in the revised version.

5. The networking analysis for co-occurrence is very interesting.

Reply: Thank you!

6. Overall, Results could be better organized to more effectively show process and information

Reply: See also our comment to reviewer #2. We also now add several points to the results as suggested.

Discussion

1. Discussion is rushed and should be better organized; discussion on totality of data, not just *Klebsiella* and *E. coli*.

Reply: Although the detection of ESBL-producing *E. coli* and *K. pneumoniae* is certainly one major finding, we now also discuss the microbiome of pangolins in more detail. See also our reply to point 3 (“importance, introduction”).

We hope, that we now adequately addressed the reviewers' concerns. On behalf of all co-authors, I thank all reviewers for taking the time to evaluate our manuscript.

Frieder Schaumburg

June 5, 2023

Dr. Frieder Schaumburg
Westfälische Wilhelms-Universität Münster
Institute for Medical Microbiology
Domagkstr. 10
Münster, North-Rhine Westphalia 48149
Germany

Re: Spectrum00664-23R1 (Pharyngeal communities and antimicrobial resistances in Pangolins, Gabon)

Dear Dr. Frieder Schaumburg:

Your manuscript has been accepted, and I am forwarding it to the ASM Journals Department for publication. You will be notified when your proofs are ready to be viewed.

Sincerely,

Katharina Schaufler
Editor, Microbiology Spectrum
